# Programming and Debugging with Semantically Lifted States

Eduard Kamburjan, Vidar Norstein Klungre, Rudolf Schlatte,
Einar Broch Johnsen, and Martin Giese

Department of Informatics, University of Oslo, Oslo, Norway
{eduard,vidarkl,rudi,einarj,martingi}@ifi.uio.no

**Abstract.** We propose a novel integration of programming languages with semantic technologies. We create a semantic reflection mechanism by a direct mapping from program states to RDF knowledge graphs. This mechanism enables several promising novel applications including the use of semantic technology, including reasoning, for debugging and validating the sanity of program states, and integration with external knowledge graphs. Additionally, by making the knowledge graph accessible from the program, method implementations can refer to state semantics rather than objects, establishing a deep integration between programs and semantics. This allows the programmer to use domain knowledge formalized as, e.g., an ontology directly in the program's control flow. We formalize this integration by defining a core object based programming language that incorporates these features. A prototypical interpreter is available for download.

## 1 Introduction

Knowledge graphs and ontologies are eminently useful representations of formal knowledge about individuals and universals. They are less suitable for the representation of change, and in particular dynamic behavior. Different approaches have been proposed that attempt to express program behavior in terms of actions on a DL interpretation [26] or a DL knowledge base [1]. A recent approach has combined a guarded command language with DL reasoning [6] to enable probabilistic model checking over the combination. Each of these approaches entails its own set of technical challenges, and they are also quite different from current state-of-the-art programming paradigms.

In this work, we present a comparatively simple connection between programs and knowledge graphs: we give a direct mapping of program states in an object-based language to an RDF graph, including the running program's objects, fields, and call stack. An immediate application of this mapping is to use semantic technology for program debugging, visualisation, querying, validation, reasoning, etc. In this paper, *semantic debugging* refers to the process of detecting and correcting conceptual mistakes at the appropriate level of global object access instead of interacting with only individual objects, as in conventional debuggers.

The RDF graph can be exposed within the programming language, which adds a semantic reflection layer to the programs. This enables *semantic programming* where the semantic view of the state can be exploited in the program. In particular, it allows one to use the knowledge of the application domain within the program. The programming language can further support a mechanism that extends the RDF graph with triples based on the results of specially designated methods. The approach is implemented and available.

In this work, the RDF graph is close to the object structure, thus representing an *internal domain* of the program. Our long-term goal is to use the same principles for *external domains* that correspond to a program's application domain. This will enable semantic integration between programs as well as linking them to external knowledge graphs and other data sources, thus extending ontology-based data access and integration (see e.g. [9]) to programs and behavioural specifications [13] that allow modeling of complex concurrent systems [16].

*Contributions.* The contributions of this paper are (1) the concept of interpreting a program state as a knowledge graph to enable semantic state access, (2) the application enabled by this concepts and (3) the SMOL language that implements the concept and exemplifies the applications.

*Paper overview.* Sec. 2 gives a motivating example of mapping program states to an 'internal' knowledge graph. Sec. 3 gives syntax and semantics of `SMOL`, the used programming language, and defines the mapping of `SMOL` program states to RDF graphs. Sec. 4 extends `SMOL` with a statement to access its own knowledge graph, thereby adding reflection by semantic state access. Sec. 5 extends `SMOL` with a way to add *computed* triples to the knowledge graph. Sec. 6 describes the implementation, Sec. 7 discusses related work, and Sec. 8 concludes. Further details of this work may be found in an accompanying technical report [17].

## 2   Motivating Example

We start by giving an example of the use of domain models in programming to demonstrate semantic bugs and motivate semantic programming. Semantic bugs are programming errors that arise from mismatches between the implementation and the domain that is implemented. Such bugs are unlikely to cause immediate runtime errors and are, thus, harder to catch. Instead, they require to examine the program state through a conceptual lens, i.e., in terms of the implemented domain. *Semantic debugging* is the process of detecting and fixing such errors.

The implemented domain can be external or internal. An external domain relates the implementation to some concept outside the program. For example, a class modeling a car that contains a list of its wheels. If the list contains only one element, this is a semantic bug: the object does not represent a car. An internal domain relates the implementation to itself.

*Example 1.* Consider 2-3 trees [2], a data structure to access key-value pairs. Such trees have three kinds of nodes: leaves have one or two data values and no children, 2-nodes that have one data value and two children and 3-nodes that

```
1 class Node(dataL, dataR, childL, childM, childR, parent)
2  get(k)
3   if(this.dataL = null)    then r := null; return r; end
4   if(k = this.dataL.key)   then r := this.dataL.value; return r; end
5   if(k <= this.dataL.key)  then r := this.childL.get(k); return r; end
6   if(this.dataR = null)    then r := this.childM.get(k); return r; end
7    ...
8   end
9 end
```

**Listing 1.** Part of the implementation of 2-3 trees in SMOL.

have two data values and three children. The keys are sorted within the tree; e.g., in a 2-node, all keys below the left child are smaller than the key stored in the node and all below the right child are larger. Lst. 2 shows an implementation of such a tree. In the implementation, a node is a 2-node if the fields `dataR` and `childR` are set to `null` and the other fields are non-null. A node is a 3-node if all fields are non-null. A leaf has no children and at least one data value.

The nodes of a 2-3 tree change their "class" during their lifetime from leaf to 2-node to 3-node. It would be highly inefficient to create a new object every time this happens, as the addition of a value to the tree may cause several such changes. This exemplifies how the application "domain" (here: 2-3 trees) and the needs of the implementing language (here: efficiency) can collide.

In this example, the distinction between leaf nodes, 2-nodes and 3-nodes is *semantic* in the sense that these nodes share the same syntactic structure. It is not visible in the code that certain nodes are not supposed to exist; e.g., nodes where only `dataR` is set to `null` are so-called *faulty nodes*. This means that erroneous insertion algorithms may cause *semantic bugs*: they may violate the domain model of 2-3 trees. Note that faulty nodes as discussed above cannot be described by means of an ontology for the syntax — the domain model is a model of runtime states (i.e., the three kinds of nodes), not one of syntax.

An interpretation of the runtime state as a knowledge graph opens for *semantic state access*; i.e., the runtime state can be queried by means of semantic tools via this interpretation. In our example, all three kinds of node can be described as both SHACL shapes and SPARQL queries for debugging.

Semantic bugs may cause *delayed* and *non-local* runtime errors. Semantic debugging requires to perform semantic queries on a program state to access the complete program state, not only the current stack trace. The faulty node described above causes the tree to effectively ignore the values below `childR` by the `get` method. Thus, the error is delayed (observable only after the faulty insertion) and non-local (not observable with a single stack trace). Similarly, the car with a single wheel may not cause *any* runtime error at all.

*Semantic programming* is the application of this conceptual lens from *within* the program: As we interpret every runtime state as a model for a domain, we

can also perform queries automatically during execution, for example, querying for all faulty nodes and calling a repair function that fixes the tree.

*Terminology.* There are some conflicts of terminology between programming languages and semantic technologies, of which we emphasise the following: In programming languages the word *semantics* describes the runtime behavior of a program, and is unrelated to ontologies and formalized domains.

In programming, a *class* is completely described by (a) its name, (b) its fields and methods and (c) the name of its superclass. We only consider class hierarchies without multiple inheritance, which form a tree. Each *object* has an identifier and belongs explicitly to one class (and its superclasses). The state, i.e., the values in the fields of an object, may change, but the class does *not*. In contrast, an *OWL class* corresponds to a unary predicate in logic and any *resource* belongs to many classes. Classes are identified by their extension and not their name, and resources have no built-in notion of class membership.

## 3    Core Language SMOL

As our programming model, we consider a semantic minimal object language (SMOL). The language contains the minimal set of features to demonstrate semantic state access: a class system to define objects and a simple while language for statements. To demonstrate the use of reflection, SMOL uses dynamic typing: each value is tagged with its type at runtime.

### 3.1   Programming Model

A program in SMOL consists of a set of classes and a **main** block. Statements and expressions are standard, including a null reference and the self reference this. For simplicity all fields are public, fields are always prefixed with the target object and nested object creation and method calls inside expressions are not supported. The syntax of SMOL is given by the following definition.

**Definition 1 (Surface Syntax).** *Let $v$ range over variables, $f$ over fields, $n$ over $\mathbb{N}$, $m$ over method names and $C$ over class names. The syntax of SMOL is defined below. We assume standard literals and operators in expressions. The notation $\overline{\cdot}$ denotes lists and $[\cdot]$ optional elements.*

Prog ::= $\overline{\text{Class}}$ **main** s **end**       Class ::= **class** C $(\overline{f})$ $\overline{\text{Met}}$ **end**       Met ::= $m(\overline{v})$ s **end**

   s ::= l:=e; $\mid$ $[l:=]$se; $\mid$ **if** e **then** s$[$**else** s$]$**end** s $\mid$ s s $\mid$ **while** e **do** s **end** s $\mid$ **return** e; $\mid$ **skip**

   se ::= **new** C$(\overline{e})$ $\mid$ e.m$(\overline{e})$       e ::= null $\mid$ l $\mid$ n $\mid$ e + e $\mid$ e $\geq$ e $\mid$ ...       l ::= this.f $\mid$ e.f $\mid$ v

The runtime semantics of SMOL is a transition system between runtime configurations. Each such configuration represents the state of the program at a given point of execution. An expression evaluates to a *domain element* (DE), which is either a literal value or an object reference. For method calls, we use runtime statements rs which extend statements s with a special statement $l \leftarrow \text{stack}$ (explained below). Runtime configurations are defined as follows.

**Definition 2 (Configuration).** *Let* $X, Y$ *range over object identifiers,* $\sigma, \varsigma$ *over maps from variables to DEs,* $\rho$ *over maps from fields to DEs and* $i$ *over* $\mathbb{N}$. *Configurations* Conf, *objects* obs *and processes* prcs *are defined by the following:*

$$\mathsf{Conf} ::= \mathsf{CT}\ \mathsf{obs}\ \big\langle[\mathsf{prcs}]\big\rangle \qquad \mathsf{rs} ::= \mathsf{s}\ |\ \mathtt{l} \leftarrow \mathtt{stack};\ \mathsf{s}$$
$$\mathsf{obs} ::= (\mathtt{C}, \rho)_X\ |\ \mathsf{obs}\ \mathsf{obs} \qquad \mathsf{prcs} ::= (\mathtt{m}, \mathtt{X}, \mathsf{rs}, \sigma)_i\ |\ \mathsf{prcs},\ \mathsf{prcs}$$

*where* CT *maps class names to a list of field names and a set of method entries. These are accessed by the auxiliary functions* fields(CT, C) *and* methods(CT, C), *respectively. We use* vars(CT, C, m) *to access the variables and* body(CT, C, m) *the body of a method. Terms* obs *are treated as sets.*

A runtime configuration Conf contains a set of objects and a stack of processes. An object has a unique name $X$ and contains its class name $C$ and memory $\rho$. A process has an id $i$ and contains the name $m$ of the method is executing, the object identifier $X$ to resolve `this`, a runtime statement $rs$ and a local store $\sigma$.

Observe that in a configuration, $\langle\mathsf{prcs}\rangle$ realizes a *stack* of processes corresponding to nested method calls. In a process, the statement $\mathtt{l} \leftarrow \mathtt{stack}$ denotes that location $\mathtt{l}$ waits for a return value from the next process on the stack.

**Definition 3 (Initial Configuration).** *Given a program* Prog, *the initial configuration has the form* $\mathsf{CT}_{\mathsf{Prog}}\ (\mathtt{Entry}, \emptyset)_E\ \big\langle(\mathtt{entry}, \mathtt{E}, \mathtt{s}, \emptyset)_1\big\rangle$, *where* $\mathsf{CT}_{\mathsf{Prog}}$ *is extracted as defined in Def. 2, but with an additional class* Entry *that has a single method* entry *with the statement of the main block as its body.*

The runtime semantics of SMOL is now presented as a structured operational semantics [24], i.e., a set of conditional rewrite rules which describe transitions from a runtime configuration into another. An expression evaluates to a pair $X, f$ or a variable $v$ if applied to a left-hand side and to a domain element if applied to a right-hand side. We denote by $[\![e]\!]_Y^{\sigma, \mathsf{obs}}$ the evaluation function for expressions $e$, where $Y$ is the value of `this`, $\sigma$ the local variables, and obs a set of objects (such that their memories $\rho$ may be accessed). For brevity's sake, we refrain from introducing the full runtime semantics here and refer to our technical report [17].

**Definition 4 (Transition System).** *The most important rules of the transition system are given in Fig. 1.*

The transition system is defined with two layers. A global layer that performs a step of the whole system and local layer that performs a step of a single statement. To connect the two layers, rule **(lift)** performs a step in the top-most process, using a local transition relation $\xrightarrow{\mathsf{CT}}_X$. The local transition relation considers only (a) the active statement (b) the current local memory and (c) all objects. We give three rules to illustrate this: **(af)** executes an assignment to some field. The left-hand side expression evaluates to the pair of object identifier and field name and the right-land side to a literal or reference to be stored there. The rule updates the heap of the target object and reduces the statement to **skip**. Rule **(av)** is similar if the left-hand side expression evaluates to a variable name. Rule **(new)** adds a new object after evaluating all parameter expressions.

$$\textbf{(lift)} \quad \frac{\mathtt{s}, \sigma, \mathsf{obs} \xrightarrow{\mathsf{CT}}_{\mathtt{X}} \mathtt{s''}, \sigma', \mathsf{obs}'}{\mathsf{CT} \; \mathsf{obs} \; \langle \mathsf{prcs}, (\mathtt{m}, \mathtt{X}, \mathtt{s} \; \mathtt{s'}, \sigma)_i \rangle \to \mathsf{CT} \; \mathsf{obs}' \; \langle \mathsf{prcs}, (\mathtt{m}, \mathtt{X}, \mathtt{s''} \; \mathtt{s'}, \sigma')_i \rangle}$$

$$\textbf{(af)} \quad \frac{\llbracket \mathtt{e} \rrbracket_{\mathtt{Y}}^{\sigma, \mathsf{obs} \; (\mathtt{C}, \rho)_{\mathtt{X}}} = v \qquad \llbracket \mathtt{l} \rrbracket_{\mathtt{Y}}^{\sigma, \mathsf{obs} \; (\mathtt{C}, \rho)_{\mathtt{X}}} = \mathtt{X}, \mathtt{f}}{\mathtt{l} \; \mathtt{:=} \; \mathtt{e;}, \sigma, \mathsf{obs} \; (\mathtt{C}, \rho)_{\mathtt{X}} \xrightarrow{\mathsf{CT}}_{\mathtt{X}} \textbf{skip;}, \sigma, \mathsf{obs} \; (\mathtt{C}, \rho[\mathtt{f} \mapsto v])_{\mathtt{X}}} \qquad \textbf{(av)} \quad \frac{\llbracket \mathtt{e} \rrbracket_{\mathtt{Y}}^{\sigma, \mathsf{obs}} = v \qquad \llbracket \mathtt{l} \rrbracket_{\mathtt{Y}}^{\sigma, \mathsf{obs}} = \mathtt{v}}{\mathtt{l} \; \mathtt{:=} \; \mathtt{e;}, \sigma, \mathsf{obs} \xrightarrow{\mathsf{CT}}_{\mathtt{X}} \textbf{skip;}, \sigma[\mathtt{v} \mapsto v], \mathsf{obs}}$$

$$\textbf{(new)} \quad \frac{|\bar{\mathtt{e}}| = |\mathsf{fields}(\mathsf{CT}, \mathtt{C})| \qquad \mathtt{Y} \; \text{fresh} \qquad \bigwedge_{1 \le i \le |\mathsf{fields}(\mathsf{CT},\mathtt{C})|} \rho(\mathtt{f}_i) = \llbracket \mathtt{e}_i \rrbracket_{\mathtt{Y}}^{\sigma, \mathsf{obs}}}{\mathtt{l} \; \mathtt{:=} \; \texttt{new} \; \mathtt{C(\bar{e});}, \sigma, \mathsf{obs} \xrightarrow{\mathsf{CT}}_{\mathtt{X}} \mathtt{l} \; \mathtt{:=} \; \mathtt{Y;}, \sigma, \mathsf{obs} \; (\mathtt{C}, \rho)_{\mathtt{Y}}}$$

**Fig. 1.** Selected rules of the transition system for `SMOL`.

Premises in the rules realize *dynamic checking*: mismatching parameters, `null` access and all other errors are caught at runtime. A runtime configuration for which no rule is applicable is *terminated*. A terminated runtime configuration with a non-empty stack is *stuck*. A program may get stuck if, e.g., a method is called on an object and this method is not defined in the object's class.

### 3.2 Semantic State Access

The formal definitions given in the previous section describe the global state of a program execution. The established way to examine this state in debuggers is to evaluate expressions in the top-most context and navigation of the process stack. To enable semantic state-access on the overall state without manual navigation, we map a configuration into an knowledge base using our `SMOL` domain model.

**Definition 5 (Knowledge Base).** *A knowledge base $\mathcal{K} = (\mathsf{T}, \mathsf{A})$ is a pair of a TBox $\mathsf{T}$ and an ABox $\mathsf{A}$. We represent the ABox as a set $E \times P \times E$, where each element is a triple over entities $E$ and predicates $P$. A triple $(e_1, p, e_2)$ is also written $p(e_1, e_2)$.*

Entities are, e.g., domain elements or method names. We remind the reader that literal values are domain elements. The sets of predicates and entities may overlap: a field is used as both an entity (to express that a class has a field), as well as a property (to connect an object with the value stored within this field).

The TBox consists of axioms. Some axioms are generated as part of the mapping and stem from the `SMOL` domain. Additionally axioms to reflect the application domain can be provided by the user.

**Definition 6 (`SMOL` Domain and Mapping).** *The generic `SMOL` domain model without the subdomain for statements is the OWL model pictured in Fig. 2.*

*The set of axioms defining the above model is denoted $\mathsf{I}_{SMOL}$. Given a configuration $\mathsf{Conf}$, the mapping $\mu(\mathsf{Conf})$ generates a ABox as defined in Fig. 3.*

Fields and locations have two roles: All fields are properties (datatype or object properties, depending on the type), and they are elements of `Field`. Treating

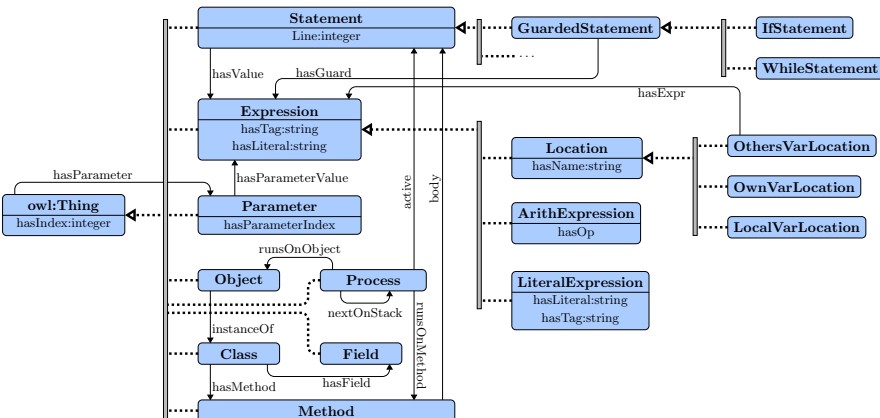

**Fig. 2.** OWL domain model for `SMOL` configurations. Common prefix `smol:` omitted. Boxes denote classes, arrows object properties and dotted arrows subclassing. Entries within the boxes are data properties.

them as properties, we can use them to connect an object `O1` with an object `O2` stored in its field `f` using the triple `f(O1, O2)`. Treating them as individuals, we can express that they belong to a class. E.g., if `O1` is of class `C`, by using the triple `hasField(C, f)`. Analogously for variables and `Location`. The domain model also structures the statements, e.g., `IfStatement` and `WhileStatement` are both subclasses of `GuardedStatement`.[1]

In the mapping, every class and object property is prefixed with an IRI unique to this version the program. We stress that each field is a subproperty of `Field` and every variable is a subproperty of `Location`. The `SMOL` domain is not merely a reformulation of the grammars in Defs. 1 and 2 — it introduces terms like guarded statements which are not given in the grammar. Exactly one state is mapped to a knowledge base at a time, not a whole trace.

*Example 2.* Fig. 4 shows a program and the main part of the mapping of its configuration. It exemplifies that the knowledge graph contains information about three layers of the program state: A syntactic layer describes the class information (in blue), an object layer describes the object instances (in brown) and a process layer describes the current stack (in yellow). The layers are not strict for fields and variables: `f` is both at the syntactic layer (`f` is a field of class `C`) and at the object layer (object `X` stores `2` in `f`) – the double line denotes equality.

The representation as a knowledge base allows us to access information beyond the basic notations of runtime semantics by inferring additional information through inference rules. We are generic in the access and inference mechanisms itself and assume some representation of axioms and queries.

---

[1] For space reasons, the complete version of this part of the domain model is given in the repository, and the following figures have been slightly simplified.

$$\mu\big(\mathtt{CT}\ \mathtt{obs}\ \langle\mathtt{prcs}\rangle\big) = \mu(\mathtt{CT}) \cup \mu(\mathtt{obs}) \cup \mu(\langle\mathtt{prcs}\rangle)$$

$$\mu(\mathtt{CT}) = \bigcup_{\mathtt{C}\in\mathbf{dom}(\mathtt{CT})} \big\{\mathtt{smol:hasField}(f_\mathtt{P}(\mathtt{C}), f_\mathtt{P}(\mathtt{f})), \mathtt{rdfs:subProperty}(f_\mathtt{P}(\mathtt{f}), \mathtt{smol:Field}) \mid \mathtt{f} \in \mathtt{CT}(\mathtt{C})\big\}$$

$$\cup \bigcup_{\mathtt{C}\in\mathbf{dom}(\mathtt{CT})} \big\{\mathtt{smol:hasMethod}(f_\mathtt{P}(\mathtt{C}), f_\mathtt{P}(\mathtt{m})), \mathtt{rdf:type}(f_\mathtt{P}(\mathtt{m}), \mathtt{smol:Method}) \mid \mathtt{m} \in \mathtt{CT}(\mathtt{C})\big\}$$

$$\cup \big\{\mathtt{rdf:type}(f_\mathtt{P}(\mathtt{C}), \mathtt{smol:Class}) \mid \mathtt{C} \in \mathbf{dom}(\mathtt{CT})\big\}$$

$$\mu(\mathtt{obs}_1\ \mathtt{obs}_2) = \mu(\mathtt{obs}_1) \cup \mu(\mathtt{obs}_2) \qquad \mu\big((\mathtt{C}, \rho)_\mathtt{X}\big) = \{\mathtt{smol:instanceOf}(f_\mathtt{R}(\mathtt{X}), f_\mathtt{P}(\mathtt{C}))\} \cup \bigcup_{\mathtt{f}\in\mathbf{dom}(\rho)} \{\mathtt{f}(\mathtt{X}, \rho(\mathtt{f})\}$$

$$\mu(\langle\rangle) = \emptyset \qquad \mu(\langle\mathtt{p}_i\rangle) = \mu(\mathtt{p}_i) \qquad \mu(\langle\mathtt{p}_i\ \mathtt{p}_j\rangle) = \{\mathtt{smol:nextOnStack}(f_\mathtt{R}(i), f_\mathtt{R}(j))\} \cup \mu(\mathtt{p}_i) \cup \mu(\mathtt{p}_j)$$

$$\mu(\langle\mathtt{prcs}\ \mathtt{p}_i\ \mathtt{p}_j\rangle) = \{\mathtt{smol:nextOnStack}(i, j)\} \cup \mu(\langle\mathtt{prcs}\ \mathtt{p}_i\rangle) \cup \mu(\mathtt{p}_j)$$

$$\mu\big((\mathtt{m}, \mathtt{X}, \mathtt{rs}, \sigma)_i\big) =$$
$$\{\mathtt{smol:runsMethod}(f_\mathtt{R}(i), f_\mathtt{P}(\mathtt{m})), \mathtt{smol:runsOnObject}(f_\mathtt{R}(i), f_\mathtt{R}(\mathtt{X})), \mathtt{rdf:type}(f_\mathtt{R}(i), \mathtt{smol:Process})\}$$
$$\cup \{\mathtt{smol:active}(f_\mathtt{R}(i), \mu(\mathtt{rs}))\} \cup \bigcup_{\mathtt{v}\in\mathbf{dom}(\sigma)} \{f_\mathtt{P}(\mathtt{v})(f_\mathtt{R}(i), \sigma(\mathtt{v}))\}$$

**Fig. 3.** Mapping configurations. Statements and expression ommited for space reasons. $f_\mathtt{P}$ adds a prefix that identifies the program, $f_\mathtt{R}$ adds a prefix that identifies the run.

```
1 class C(f) m(v) return this.f + v; end end
2 main o = new C(2); o.m(3); end
```

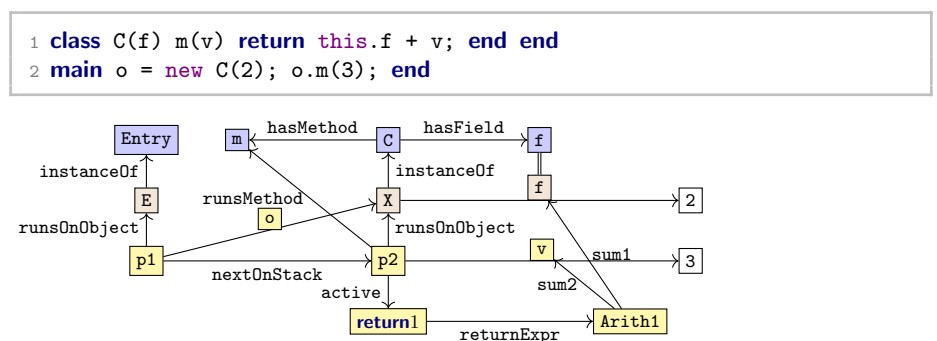

**Fig. 4.** A `SMOL` program and its mapping into a knowledge graph after `o.m(3)` is called.

**Definition 7 (Queries).** *An answering engine is a function that maps a knowledge base and a query to a set of answers:*

$$\mathsf{ans}\big((\mathsf{T}, \mathsf{A}), q\big) = \big\{\overline{x} \mid (\mathsf{T}, \mathsf{A}) \models q(\overline{x})\big\}$$

*The satisfiability relation $\models$ depends on the concrete nature of the query.*

Query engines can also handle boolean queries, such as description logic assertions, by returning either an empty set for true or a non-empty set for false. *Semantic Debugging.* The mapping allows the programmer to debug a program by simple and efficient access to the runtime configuration through queries. This does not require an axiomatisation of the application domain, but if one is available, it can be used to additionally debug in terms of the application domain.

*Example 3 (Semantic Debugging).* Continuing with Ex. 1, we can formalize the notion of leaves, 2-nodes, 3-nodes and faulty nodes with the OWL class expres-

$$
\begin{array}{ll}
\textsf{Root} & \equiv \textsf{parent} : \textsf{null} \\
\textsf{Leaf} & \equiv (\textsf{childL} : \textsf{null}) \sqcap (\textsf{childM} : \textsf{null}) \sqcap (\textsf{childR} : \textsf{null}) \sqcap (\exists \textsf{dataL}.\exists \textsf{instanceOf}.\textsf{Pair}) \\
\textsf{TwoNode} & \equiv (\textsf{childR} : \textsf{null}) \sqcap (\textsf{dataR} : \textsf{null}) \sqcap (\exists \textsf{childL}.\exists \textsf{instanceOf}.\textsf{Node}) \\
& \quad \sqcap (\exists \textsf{childM}.\exists \textsf{instanceOf}.\textsf{Node}) \sqcap (\exists \textsf{dataL}.\exists \textsf{instanceOf}.\textsf{Pair}) \\
\textsf{ThreeNode} & \equiv (\exists \textsf{dataL}.\exists \textsf{instanceOf}.\textsf{Pair}) \sqcap (\exists \textsf{dataR}.\exists \textsf{instanceOf}.\textsf{Pair}) \\
& \quad \sqcap (\exists \textsf{childL}.\exists \textsf{instanceOf}.\textsf{Node}) \sqcap (\exists \textsf{childM}.\exists \textsf{instanceOf}.\textsf{Node}) \sqcap (\exists \textsf{childR}.\exists \textsf{instanceOf}.\textsf{Node})
\end{array}
$$

**Fig. 5.** Domain model for 2-3 trees as OWL classes in DL syntax.

sions from Fig. 5. To access a configuration $\textsf{Conf}$ for semantic debugging with a query $q$, one needs to compute

$$
\textsf{ans}\big((\textsf{T}_{\textsf{SMOL}} \cup \textsf{T}_{\textbf{user}}), \mu(\textsf{Conf})), q\big)
$$

where $\textsf{T}_{\textbf{user}}$ are user input axioms that define the domain of the application.

Note that the axioms $\textsf{T}_{\textsf{SMOL}}$ and the mapping $\mu$ are part of the language and need *not* be provided by the user. The vocabulary is partially given by the language itself and partially by the user-defined classes. The overhead for the programmer is, thus, only to provide additional axioms if needed.

Recall that semantic state access is performed on a single state. For example, if the construction of the 2-3 tree in Ex. 1 temporarily contains a faulty node, then this node will not be retrieved by a semantic state access *after* the construction has completed (if the construction indeed fixes the node). If one wishes to detect such a faulty node without explicitly introducing it into the domain model, one can either use a query language with negation or a language for validation such as SHACL, whose integration into $\textsf{SMOL}$ we discuss in Sec. 6.

Semantic debugging differs from traditional debugging in several points: (1) Access is performed with a query language to examine larger states, instead of manually investigating the call and heap structure of a snapshot. (2) The ability to query and debug using application domain knowledge. This becomes critical when investigating complex data structures where the implementation and the application domain conflict, as illustrated by Ex. 1 where the domain differentiates 3 classes of nodes, but the implementation has only one.

Semantic technologies are not a standard tool for debugging, but given the complexity of some current debugging tools, such as profilers, we expect that the overhead for the programmer is acceptable for complex structures, especially if they have an application domain component.

*Verification.* Before we investigate semantic state access further, we note that using established programming language constructs for control flow allows us to directly use results from programming languages. For example, we can carry over verification techniques and verify invariants. For a more detailed treatment of the following statement, we again refer to our technical report. We say that a formula is *state-based* if it contains as predicates only (a) `instanceOf` and (b) subproperties of `Field` and `Expression`. An ontology is *state-based* if its axioms contain only such predicates.

**Proposition.** *Let* Prgm *be a* SMOL *program, such that type inference succeeds. Let $O$ be a state-based domain ontology,* C *a class in* Prgm *and $\varphi$ be a state-based description logic formula. There is a sound and complete (relative to integer arithmetic) proof system that checks whether $\varphi$ is an invariant for* C.

## 4   Internal Semantic State Access

So far, semantic state access is used to query the program state from the outside to realize semantic debugging. Next, we integrate domain knowledge directly into the runtime semantics of the program: the control flow is expressed not in terms of data structure implementations, but in terms of the implemented domain. To do so, we add a new statement to retrieve information about the state through the conceptual lens of ontologies about the program state from the *inside*.

**Definition 8 (Extended Surface Syntax).** *Let str range over string literals. The syntax of* SMOL$^+$ *is defined by extending statements from Def. 1 as follows:*

$$\texttt{s} ::= \ \dots \ | \ \texttt{l:=access}(str, \overline{\texttt{e}});$$

The runtime semantics of the access statement map the current program and use the *inferred* knowledge graph to perform some query. The query is not static: additional parameters are evaluated and mapped to nodes inside the knowledge graph. We use %$i$ as placeholders in the query string to add the additional parameters. We require the existence of a List class to represent the results of the statement for further computations.

**Definition 9.** *The runtime semantics* SMOL$^+$ *are defined by all rules from Def. 3 and the following additional rule for the access statement. We remind that* I *and* R *are available as user inputs.*

$$(\textsf{acc}) \ \frac{\texttt{q} = str\big[\%i \setminus [\![\texttt{e}_i]\!]_Y^{\sigma,\textsf{obs}}\big]_{i \leq n} \qquad l = \textsf{ans}\big(((\textsf{R},\textsf{I}), \mu(\textsf{Conf})), \texttt{q}\big)}{\textsf{CT obs} \ \underbrace{\big\langle \textsf{prcs}, \ (\texttt{m}, \texttt{Y}, \texttt{v} := \texttt{access}(str, \texttt{e}_1, \dots, \texttt{e}_n); \ \texttt{s}, \sigma)\big\rangle}_{=\textsf{Conf}} \ \to_\mathsf{T} \textsf{CT obs} \ \big\langle \textsf{prcs}, \ (\texttt{m}, \texttt{Y}, \texttt{s}, \sigma[\texttt{v} \mapsto l])\big\rangle}$$

We illustrate internal SSA ith two examples: domain-specific control and reflection. We can apply the *domain* ontology in the queries, inspired by the 'ontology-mediated symbols' introduced by Dubslaff et al. [5], to directly control the program in terms of the domain. An ontology-mediated symbol is a query over the extended knowledge graph and can be expressed in our system.[2]

*Example 4.* Consider the upper code in Lst. 4. The scheduler uses the domain knowledge to determine which of the platforms is overloaded and accordingly moves servers between platforms. The critical point here is that :Overloaded is defined by the background knowledge and can, thus, be changed according to different scenarios outside the simulation.

---

[2] Dubslaff et al. use description logic formulas for queries, not SPARQL, but our approach is general w.r.t. the query language.

```
1 class Platform(serverList) ... end   class Server(taskList) ... end
2 class Scheduler(platformList)
3   reschedule()
4     over := access("SELECT ?x WHERE{?x a :Overloaded }");
5     tasks := this.collectExcessiveTasks(over);
6     this.reschedule(tasks);
7   end
8 end
```

```
1 m(o)
2  callable :=
3   access("SELECT ?y WHERE{%1 :instanceOf ?y. ?y :hasMethod n }",o);
4  if callable <> null then o.n();   else .../*report error*/ end
5 end
```

**Listing 2.** Upper code: Using ontology-mediated symbols in `SMOL`. Lower code: Reflection with semantic state access.

Exchanging the background knowledge can be used to change the specification when a platform is overloaded — the language concepts of ontology-mediated programming are thus subsumed by semantic programming.

The class table and stack structure are both available in the knowledge graph. We can, thus, reason about these structures at runtime *in terms of the formal (runtime) semantics and domain*. I.e., this does not merely expose the structure of the implementing runtime environment but adds domain knowledge — in this case, the domain of runtime configurations. The actual implementation in the interpreter, or other runtime, may for efficiency be quite different.

*Example 5.* The method in the lower part of Lst. 4 checks that a passed parameter is from a class that implements `n` before calling the method.

The above example illustrates a common pattern in languages with dynamic types or reflection and is based on the static information. As the knowledge graph underlying the state also enables access to the processes, one can also use it to make control based on the stack. E.g., one can bound recursion without an additional counter or reflect on the calling method without passing a parameter.

## 5   Computational Semantic State Access

So far, we can access the data in a configuration and, beyond merely serialising it into another format, use inference to ontologise it. However, we cannot access data that is implicit in the configuration. Consider Let. 5: The `Rectangle` class has a width `w` and a height `h`, but its area is not directly available. Even worse: each rectangle is part of a scene that may scale all its elements (and apply further operations, which we omit here). While the final step of the computation of an area is a multiplication, the overall computation involves a method call.

Our solution is *computational semantic state access* (CSSA): certain methods, here `area` are directly encoded as inference rules to enrich the knowledge

```
1 class Scene(scaling) getScale() return this.scaling; end end
2 class Rectangle(scene, w, h)
3  rule area() s := scene.getScale(); return s*this.w*this.h; end
4 end
5 main sc := new Scene(2); ... r := new Rectangle(sc, 5, 1); end
```

**Listing 3.** A rectangle inside a scene.

graph. This makes data that is a *computational* result available for inference. Furthermore, this allows one to determine based on a query where a computation has to be performed (instead of pre-performing it on all possible targets).

The language allows the programmer to mark methods as available for inference by exposing them with the `rule` keyword.

**Definition 10 (Extended Surface Syntax with CSSA).** *The syntax of Def. 8 is extended by replacing the method definition (from Def. 1) with*

$$\mathsf{Met} ::= [\mathtt{rule}] \; \mathtt{m}(\overline{\mathtt{v}}) \; \mathtt{s} \; \mathbf{end}$$

For wellformedness, we demand that `m` is guaranteed to terminate if it is modified by `rule`.[3] The semantics of a `rule` method is not a transition rule, but an extension of the translation.

**Definition 11.** *The semantics of* $\mathsf{SMOL}^+$ *is defined by the original transition system of* $\mathsf{SMOL}^+$ *and by replacing the definition of* $\mu((\mathtt{C}, \rho)_\mathtt{X})$ *in Fig. 3 by*

$$\mu\big((\mathtt{C}, \rho)_\mathtt{X}\big) = \{\mathtt{smol} : \mathtt{instanceOf}(f_{\mathtt{run}}(\mathtt{X}), f_{\mathtt{prog}}(\mathtt{C}))\} \cup$$
$$\bigcup_{\mathtt{f} \in \mathbf{dom}(\rho)} \{f_{\mathtt{prog}}(\mathtt{f})(f_{\mathtt{run}}(\mathtt{X}), \rho(\mathtt{f}))\} \cup \bigcup_{\substack{\mathtt{m} \; is \; a \\ \mathtt{rule} \; of \; \mathtt{C}}} \{f_{\mathtt{prog}}(\mathtt{exec\_C\_m})(f_{\mathtt{run}}(\mathtt{X}), l)\}$$

*where $l$ is a literal computed as follows. Let $\mathcal{K}$ be a knowledge graph and $\mu_{obj}^{-1}(\mathcal{G}) = (\mathsf{CT} \; \mathtt{obs} \; \epsilon)$ its state without the processes. Let $\mathtt{X}$ be the object id of the object bound to $?o$. The configuration $\mathsf{CT} \; \mathtt{obs} \; \big\langle (\mathtt{m}, \mathtt{X}, \mathtt{s}, \{\})_1 \big\rangle$ finishes in a configuration of the form $\mathsf{CT} \; \mathtt{obs}' \; \big\langle (\mathtt{m}, \mathtt{X}, \mathbf{return} \; \mathtt{e}, \sigma)_1 \big\rangle$. The literal $l$ is defined as the evaluation of $\mathtt{e}$ in this configuration. If the execution does not finish in a configuration of the required form, we set $l = \mathtt{smol:null}$.*

The execution is not performed on $\mu(\mathsf{Conf})$ itself. This means that any state change, e.g., object creations or changes of fields, are *not* recorded in $\mu(\mathsf{Conf})$. We stress that the `access` statement is still part of the language.

*Example 6.* Consider the final configuration of the program in Lst. 5. Let $\mathtt{X}$ be the created object. The method `area` is syntactically guaranteed to terminate and application to $\mathtt{X}$ results in the following added triple:

$$\mathtt{prog:exec\_Rectangle\_area}(\mathtt{run} : \mathtt{X}, 10)$$

---

[3] Via a timeout, a syntactic check that no (mutual) recursion and no loops occur during its execution, or a termination proof. We do not commit to a concrete restriction.

We remind the reader that this is not merely an arithmetic expression, but requires a method call to include the scaling of the overall scene— `rule`-methods allow one to include such computations directly into the knowledge graph.

A method exposed with `rule` takes no parameters to avoid spurious triples. If a call with a particular parameter is required, one may introduce a wrapper class that is created before $\mu(\mathsf{Conf})$ is computed.

*Example 7.* To lookup 5 in Ex. 1 in a query, one may introduce the following class and create an instance `new Wrap(tttree, 5)` before a `get` call.

```
1 class Wrap(t, key) rule lookup() v := t.get(key); return v; end end
```

CSSA is not redundant to method calls: it extends the knowledge base at *every* instance and allows to select in the graph depending on these attributes.

## 6  Implementation

An implementation is available at `github.com/Edkamb/SemanticObjects`. It supports a superset of $\mathsf{SMOL}^+$ and adds inheritance and some convenience features for the program, such as output. Several examples are provided, including the example from Sec. 2 with complete, and a $\mathsf{SMOL}^+$ implementation of a subset of the geological assistant [4], a simulator for geological processes.

*Interpreter.* $\mathsf{SMOL}^+$ is implemented by an interpreter written in Kotlin that builds on Apache Jena and HermiT [10] for the semantic state access. The interpreter implements an interactive shell to realize a Read-Evaluate-Print-Loop (REPL) that allows the user to step through the execution and semantically access the current state. The user may query the state with `SPARQL`, validate it against `SHACL` shapes or retrieve all members of an `OWL` class defined by a class expression. As a domain ontology, a file containing `OWL` classes can be loaded. It is possible to run a program without stepping through the execution; the language is extended with a `breakpoint` statement that stops the execution in this case. Additionally, the interpreter uses the prefixes `prog:` (for $f_{\mathsf{P}}$) and `run:` (for $f_{\mathsf{R}}$) to simplify referencing elements of the current state. CSSA is implemented by introducing a Jena-functor for each `rule` method and a rule with this functor in its head. The functor copies the complete interpreter state without the stack and executes the corresponding method.

*Performance.* We have evaluated the performance of internal SSA by adding $n$ elements to a 2-3 tree and the querying OWL class expression using HermiT. For $n{=}100$, the system used 150s. To evaluate CSSA, we have similarly added $n$ elements and then used the `rule` wrapper from Ex. 7 to retrieve a value. Here, for $n{=}3000$, the system used 205s. 3000 added values correspond to approximately 9000 created `SMOL` objects[4] and 60k triples. This shows that our proof-of-concept can handle non-trivial amounts of data, and complex data structures and queries.

---

[4] Partly because the implementation still lacks a garbage collector for rewritten nodes.

We conjecture that the most significant bottleneck concerning the performance is the non-optimised interpreter, and the explicit generation of the knowledge base. To increase efficiency, we plan to make the knowledge base virtual and only access object states as required to answer queries. Backward reasoning can be included e.g. by query rewriting for OWL QL ontologies.

## 7   Related Work

Ontologies for Java's core concepts by Kouneli et al. [18] and for connecting object-oriented languages by de Aguiar et al. [3] have similarities to SMOL's ontology, but aim at communication between users and not at semantic state access.

Imperative programming languages and transition systems can operate directly on knowledge graphs through atomic actions. Golog [22] uses first-order logic guards to examine and pick elements from its own state. *knowledge-based* programs [8] support an epistemic knowledge modality $K$. Zarrieß [26] integrates description logic in a concurrent extension of Golog to verify CTL properties with description logic assertions. These assertions are easily realised using assert in SMOL$^+$, while our object invariants are orthogonal to CTL checking of traces. When operating on a knowledge graph, an ABox may change and violate a TBox. Calvanese et al. [1] propose two operations ASK and TELL for transition systems defined *explicitly* over knowledge bases. ASK corresponds roughly to our access, while TELL performs an action required by the explicit representation. In contrast, the transition system in SMOL$^+$ is *implicit* such that well-established principles from programming languages carry over to avoid reinvestigations of modularity, runtime semantic structure and control flow for knowledge bases. While all changes to the knowledge graph are global in Calvanese et al., global changes in SMOL$^+$ only happen in the part of the knowledge graph inferred from user-provided axioms; the part inferred from the mapping only changes locally.

Our work has not investigated programming languages that operate directly on DL interpretations or knowledge graphs, as done by [1,26], but rather how programming languages can be enhanced by semantic technologies. Closest to our work, *ontology-mediated* programming [5,6] defines an interface to integrate additional knowledge into a stochastic model checking tool, using external knowledge graphs to influence control flow. In contrast, we use internal knowledge graphsfor debugging. Neither the application to debugging nor an integration with rule-based inference as in CSSA has been studied in any of these approaches. Additionally, these approaches all use unconventional operators or highly specialised paradigms, while SMOL$^+$ allows external semantic state access for a standard object-oriented language; the access and rule extensions are optional and based on a clear interface and established query language, instead of low-level logic-based operations. For these reasons, SMOL$^+$ appears as conceptually simpler.

Ontologies can be used to type programs. Leinberger et al. [19] study DL concept expressions as static types in a $\lambda$-calculus, and type check using SHACL constraints [21]. Existing programming languages can be integrated with RDF data using the type systems of Paar and Vrandecic [23] and Leinberger et al. [20].

The difference between ontologies and regular types is not merely one of taste: (a) concepts allow more expressive structure than type hierarchies and (b) classes in programming languages are designed by the user to fit the needs of its application, while the concepts of the domain are designed to accomodate the needs of a general domain. While this work attempts to unify two tools made for different tasks, our approach is to give a sensible interface. `SMOL` is dynamically typed and the concepts of the domain and mapping in `SMOL` are disjoint and need to be connected using additional axioms. The connection to types has also been investigated through mappings [15] and code generation [25].

Eiter et al. [7] explore answer set programming to embed rules over DL knowledge graphs in the declarative setting of logic programs. Their rules are more expressive than our CSSA rules and aim to be a general programming approach. Käfer and Harth [14] perform actions on RDF files in the semantic web using linked data, operating on a set of user-input rules for an abstract state machine. Horne et al. define an operational semantics for SPARQL updates [12] and a system that internalizes queries into a process algebra [11].

## 8    Conclusion

This paper presents a novel approach to combine semantic technologies and programming languages. By regarding runtime configurations as knowledge graphs, we can use semantic state access to query such configurations for semantic debugging. By adding a semantic reflection layer to the programming language, computations can be driven by the result of queries from within a program. Finally, a deep integration of inference and computation allows inference to trigger method executions through computational semantic state access.

*Future Work.* As discussed, we plan to make the knowledge base virtual for performance reasons. We are also considering to develop an extension with special statements to manipulate the TBox and investigate how further programming languages concepts, such as garbage collection and encapsulation, carry over.

**Acknowledgements**  We thank Clemens Dubslaff and Patrick Koopmann for inspiring discussions on ontology-mediated verification. We are grateful to the anonymous reviewers for very constructive comments. This work was supported by the Research Council of Norway via *SIRIUS* (237898) and *PeTWIN* (294600).

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
