# OpenReview forum: "Programming and Debugging with Semantically Lifted States"
_eswc-conferences.org/ESWC/2021/Conference/Research_Track — ESWC 2021 Research_

### Official Review · AnonReviewer1 · 2021-01-04
**Intriguing approach, I am not sure that it is ready for prime time**

**Rating:** 1
**Confidence:** 4
**Impact:** 2
**Design And Technical Quality:** 4

**Review:**

In the following I share my thoughts that I had as I read the paper from beginning to end. The synthesis of these thoughts is given in the weak and strong points.

Overall, I decide in favor of the paper because of its originality, though I am on the fence about the clarity and significance of the paper.

Introduction:

The paper does not clearly position itself wrt [1,2,3]. While I am inclined to believe that this submission addresses a different objective than  [1,2,3],
this submission does not clearly say what its objectives are. "we present a comparatively simple connection" is a "how" and not an objective. The possible objective of "semantic debugging" is unclear, as it does not define what is meant. I did not understand the sentence "at the more appropriate level of global object access...."

One might have the impression that "semantic debugging" is related to type checking using knowledge graph backgrounds, such as done in

Martin Leinberger, Philipp Seifer, Claudia Schon, Ralf Lämmel, Steffen Staab:
Type Checking Program Code Using SHACL. ISWC (1) 2019: 399-417

Martin Leinberger, Ralf Lämmel, Steffen Staab:
The Essence of Functional Programming on Semantic Data. ESOP 2017: 750-776

- how does this submission compare to these papers, as these papers also allow "one to use the knowledge of the application domain within the program".



Motivating Example

This section makes the notion of "semantic debugging" a bit clearer. However, this particular example looks a lot as if it could by verification applied to abstract data types. How does the paper position itself with such related work?
Furthermore, I am not convinced that one should query for faulty nodes and call a repair function. I would rather want to have a correct implementation of this class.


Programming Model
gave me a flavor of the language. I did not succeed in completely understanding the transition system.

Semantic State Access
I understand the idea to query for configurations. I see  issue for which I do not understand the answer:

(i) An implementation that constructs a 2-3-node may temporarily have an invalid configuration. Comparable to a database transaction, this is fine and sound, if at the end of a construct method the node has the right properties. Would a query for a wrong configuration not possibly lead to many such "false positives" that appear during the construction of such a node.

(ii) The ontology in Fig 6 describes possible "correct" configurations. How would one ask for "wrong configurations". Would we have to manually query for all possibilities (or the negation of the union). Would this be practically feasible / how tractable would this be for a large number of allowed configurations?

(iii) the user will have to understand the structures that are illustrated in Figure 5. Will these not be much harder to understand than possible semantic mistakes in the first place?

Internal semantic state access AND Computational semantic state access
While possibly useful, I find that these sections add further bells and whistles before the paper has clearly established the usefulness of proceeding sections. From a paper reading perspective, I would favor the description of an extended use case that illustrates applicability of the core concepts before the addition of these further capabilities. In particular, I do not understand thus far, how - assuming I had written a program of several dozen lines of code - I would approach the problem of semantically debugging this program.





**Anonymity:**

Yes, I would like my review to remain anonymous.

**Reuse And Availability:**

4: High

**Strong Points:**

- The approach is highly original
- I find the idea of the paper to semantically access program states intriguing
- The evaluation gives some insight into the scalability of the approach

**Subreviewer:**

I submitted this review.

**Weak Points:**

- I have not understood from the paper whether the proposed approach would be practically useful for programmers or whether the burden put on the programmer by the approach would be higher than the challenge of identifying bugs without the approach. Also, I am back and forth about when I would actually try to use such an approach assuming that its use would be intuitive (Which I do not think is the case now).

- The paper is weak on relating its contribution to a range of other approaches, especially in programming languages. Note that I expect not only a comparison to work that explicitly uses semantic technologies, but also to established work in programming languages that pursues similar goals. Abstract data types or programming by contract come to my mind (I am not an expert in PL).

---

> ### Author Rebuttal · Authors · 2021-01-29
>
> # Answer to AnonReviewer 1
> We thank the reviewer for the very constructive and thorough comments.
>
> When revising the paper, we are going to address the raised points as follows.
> As the additional discussion of related work may take up some place, we plan to publish a technical report that contains the full version of 3.1 and remove the transition system from the conference publication, if necessary.
>
> # Specific answers:
> - to (i):   SSA considers a single state, so the "invalid" nodes are not available at the end of the construction
> - to (ii):  Querying for "invalid" configurations using SPARQL or a similar language indeed requires a negation. However, we use a more general notion of query and also implement SHACL validation that is more suited to this task.
> - to (iii):
> We are motivated by our experiences with formalizing evolving systems and developing simulators (such as Railway systems [6]) where the resulting applications duplicate domain knowledge in both data structures and behavior.
> For such applications, which are very near to their domain, it is difficult to (a) spot semantic bugs, which are only recognized as bugs by domain experts
> (b) reduce redundancy w.r.t. established ontologies, as the program must not only mirror the application domain, but also follow established practices to be well-simulatable.
> Semantic programming and debugging allow us to solve these problems.
>
> As mentioned in the article, we have re-implemented one such system for geological processes [11] in SMOL. We will add a more detailed discussion on such systems to the introduction. We hesitate to fully include this case study, as introducing the geological domain seems beyond the scope of this paper and would obscure the conceptual contribution.
>
> We want to point out that semantic debugging and semantic programming target specific classes of programs which must incorporate application domain knowledge:
>
> Semantic programming allows one to express control flow in terms of *domain knowledge* from the *application domain* (see Ex. 4),
> while semantic debugging plays a two-fold role:
> (a) It is critical to debug semantic programs (i.e., programs using SSA) and to debug programs in terms of their application domain. E.g., to debug a geological simulator in terms of the geological domain. Such debugging is not supported by current debuggers.
>
> (b) It *also* enables the investigation of complex data structures where the implementation and the application domain conflict, as illustrated by Ex. 1:
> here the domain differs between 3 classes of nodes, but the implementation has only one (because a node may change from a 2-node to a 3-node and there is a high performance penalty in creating a new object).
> Given the complexity of some current debugging tools, such as profilers, we are positive that the overhead for the programmer is acceptable for more complex structures, especially if they have an application domain component.
>
> We will discuss these points together with the discussion of targeted applications.
>
>
> ## General point on related work and contribution:
>
> The contribution of this work is three-fold (1) the concept of interpreting a program state as a knowledge graph to enable semantic state access (2) the application enabled by this concepts and (3) the SMOL language that implements the concept and exemplifies the applications. We will clarify this better in the introduction.
>
> We will make our position w.r.t. to [1-3] more explicit:
> Contrary to [1] and [2] we do not investigate programming languages that operate directly on DL interpretations or knowledge graphs, but
> investigate how programming languages can be enhanced with semantic technologies. Contrary to [3] we do so not on external knowledge graphs, but on an internal one.
>
> Note that we do not consider SSA an alternative to types, but as complementary to them.
>
> We thank the reviewer for their suggestion to also discuss related work in a broader PL context and for the concrete pointers to additional related papers. We agree with these remarks and will add a separate section on related work in the revised version of the paper

---

### Official Review · AnonReviewer4 · 2021-01-13
**interesting paper with novel work but limited positioning with respect to SotA**

**Rating:** 1
**Confidence:** 2
**Impact:** 3
**Design And Technical Quality:** 3

**Review:**

This is a paper related to programming and debugging with semantically lifted states. The paper proposes a mapping of program states in an object-based language to an RDF graph to achieve semantic debugging. Semantic debugging is needed to detect bugs that do not cause immediate runtime errors and it is thus hard to catch because domain knowledge is required.

Overall, the paper is well-written. The solution is sound and its explanation clear, even though certain aspects could have been better clarified. To the best of my knowledge, the work is very novel and it deserves attention.

I include a couple of more detailed comments for clarifications:

- In the paper it is mentioned "Each of these approaches entails its own set of technical challenges, and they are also quite different from current state-of-the-art programming paradigms." but it is not explained which are the challenges of current approaches and which their weaknesses that lead to the proposed approach that is supposed to be "simpler".

- I think the following statement is a bit confusing: "Field and Location are both object properties and classes. ...  Each field is a subproperty of Field and every variable is a subproperty of Location.", but does this hold only in the case Field is a property? otherwise how can this hold if Field is class?

- "The direct mapping allows the programmer to debug a program by simple and efficient access to the runtime configuration through queries. This does not even require an ontology, but if one is available, it can additionally be used to debug in terms of the application domain."  This statement left me skeptical and I would appreciate a clarification regarding the role of semantics

- Example 2 and Figure 5 refer to syntactical, object and process layer but these are not evident in the figure

- spelling mistake: "An ontology is state based if if uses only such predicates in any of its axioms." change to "if it uses"

- Figures including code extracts could have been Listings

after rebuttal
------------------
I would like to thank the authors for answering to my comments.

**Anonymity:**

Yes, I would like my review to remain anonymous.

**Reuse And Availability:**

3: Medium

**Strong Points:**

- The paper covers a **very novel topic** where semantics are applied
- The paper is **well-written** overall

**Subreviewer:**

I submitted this review.

**Weak Points:**

- There is **very limited related work** presented. The related work is very limited, to its greater extend focuses on a single paper (reference [2] of the paper). I am not sure though if there is much more related work, especially on applying semantics to program states.

- Even though there is some relevant work, the proposed approach is **not compared to existing work**.  The paper only provides a very high level description of the differences between the proposed approach and the approach proposed at [2]. No systematic comparison is performed neither on the semantic model production/consumption nor on the implementation. (I assume the later occurs because [2] does not seem to provide an implementation, so such a comparison would not be feasible)

- Even though the paper is well-written, sometimes it becomes hard to follow, mostly because certain aspects of the solution do not come with enough explanation

- The potential **impact of the work is not well-highlighted**

---

> ### Author Rebuttal · Authors · 2021-01-29
>
> # Answer to AnonReviewer 4
> We thank the reviewer for the very constructive and thorough comments.
>
> ## Specific points:
>  - On the statement on Fields and Locations:
> All fields are subclasses and subproperties of Field. Analogously for variables.
> By being properties, we can express that to use them to connect an object O1 with the object O2 stored in its field F using the triple F(O1, O2).
> By being objects, we can express that they belong to a class. E.g., if O1 is of class C using the triple hasField(C, F).
> This is also the reason the f occurs both over an arrow (from X to 2) and at the end of one in the figure (hasField).
>
> We are additionally going to mark the nodes in Fig. 5 according to their layer (syntactical/object/process) and rearrange the labels.
>
>  - On the role of semantics:
> "This does not even require an ontology." means that semantic debugging can be performed with an ontology for the application domain.
> The ontology for SMOL states is of course always required.
> We are going to change this sentence to "This does not require an axiomatisation of the application domain.".
>
>
>
> ## General point on related Work and "simpleness":
> We will make our position w.r.t. to [1-3] more explicit:
> Contrary to [1] and [2] we do not investigate programming languages that operate directly on DL interpretations or knowledge graphs, but
> investigate how programming languages can be enhanced with semantic technologies. Contrary to [3] we do so not on external knowledge graphs, but on an internal one.
>
> We believe that our approach is simpler as it reuses the established structure of OO programs with a clear query-language based interface to the ontological part.
>
> We thank the reviewer for the suggestion to better clarify the differences to prior approaches. We agree with these remarks and will add a separate section on related work in the final version of the paper. As the additional discussion of related work may take up some place, we plan to publish a technical report that contains the full version of 3.1 and remove the transition system from the conference publication, if necessary.

---

### Official Review · AnonReviewer2 · 2021-01-14
**Programming and Debugging with Semantically Lifted States**

**Rating:** 1
**Confidence:** 4
**Impact:** 3
**Design And Technical Quality:** 3

**Review:**

This paper is about representing  the internal state of the execution of a program using Semantic Web language RDF and querying this representation using SPARQL.
In addition, it is possible to provide additional  Description Logics (DL) statements that define the semantics of program objects themselves represented as RDF.

A toy language, SMOL, is used in the paper. The semantics of SMOL is given using semantics inference rules. This part is difficult to grasp and I suspect that  SW readers will have difficulty with this. In particular the l <- stack statement must be explained.

Some questions remain :

Is the approach generalisable to real programming languages ?
Is the approach scalable ?
What is the added value w.r.t usual programming environment debuggers ?


I acknowledge the rebuttal.

**Anonymity:**

Yes, I would like my review to remain anonymous.

**Reuse And Availability:**

2: Low

**Strong Points:**

Interesting conceptual idea


**Subreviewer:**

I submitted this review.

**Weak Points:**

Semantics part much to complex w.r.t. the purpose of the paper.
The approach is implemented on a specific SMOL toy language.

---

> ### Author Rebuttal · Authors · 2021-01-29
>
> #Answer to AnonReviewer 2:
> We thank the reviewer for the very constructive and thorough comments.
>
> We expect the proposed approach to scale both to bigger states (as discussed, we are investigating virtual graphs for this purpose) and larger programming languages. Concerning larger programming languages, we expect that applying SSA to a more complex language will provide even more benefit, as their states are more complex and harder to investigate in detail. This is in particular the case for languages such as Java, where the JVM implementation of the language drastically differs from the surface syntax exposed to the programmer. An ontology for a mainstream language, however, is beyond the scope for this paper and would obscure the conceptual contribution.
>
> The added value w.r.t. usual programming environment debuggers is (1) access to a query language to examine larger states and (2) the ability to query and debug using application domain knowledge. This becomes critical when investigating complex data structures where the implementation and the application domain conflict, as illustrated by Ex. 1 where the domain differentiates 3 classes of nodes, but the implementation has only one (because a node may change from a 2-node to a 3-node and there can be a high performance hit to create a new object).
>
> Finally, semantic debugging is needed to debug semantic programs (i.e., programs using CSSA and internal SSA).

---

### Official Review · AnonReviewer3 · 2021-01-15
**An interesting approach**

**Confidence:** 4
**Impact:** 3
**Design And Technical Quality:** 4

**Review:**

I acknowledge the rebuttal. Looking at my and the other reviewer's reviews and the corresponding rebuttals, I see that the authors could fix the problems identified by the reviewers, but the changes would be substantial so the new paper would deserve another round of reviews, hence I stick with my negative verdict.

The paper is about a method to expose the state of the execution of a computer program as RDF, and to allow for reflection, ie. using the state information in programming, by allowing for querying this state in RDF from within program code. The authors present a programming language, SMOL, with a syntax akin to Java, a formal view on a execution's state, and a transition system with rules that defines how to move between states. Then they present a lifting of these states to RDF, and how to put the querying and reflection part into the formal definitions. The authors have a publicly available implementation. They very briefly shed light on the performance of their approach. Related work is only very briefly covered and scattered over the conclusion and introduction.

In terms of originality, I am not sure, I don't doubt that the approach is novel, yet I guess reflection is not new, to put the state of program execution in first-order structures is not new, and there exist relevant papers between semantic web and programming. A better related work section would help, and a list of contributions in the beginning (is SMOL one of the contributions of the paper or just a vehicle?).
Some authors and papers to have a look at:
* Leinberger et al.: Semantic Web Application Development with LITEQ, ISWC 2014 <-- using RDF from a programming language
* Horne: Programming languages and principles for read-write linked data. PhD thesis, University of Southampton 2011
* Stärk et al.: Java and the Java Virtual Machine: Definition, Verification, Validation. Springer 2001 <-- they use Abstract State Machines, ie. first-order structures with a set of transition rules to describe an how Java execution states evolve. There is a whole body of work around ASMs. As the authors are mentioning their relation to Java, I mention this particular work.
* Käfer and Harth: Rule-based programming of User Agents for Linked Data, LDOW 2018 <-- Abstract state machines on the web: state in RDF, transition systems in N3 rules.



**Anonymity:**

Yes, I would like my review to remain anonymous.

**Rating:**

-1: Weak Reject

**Reuse And Availability:**

4: High

**Strong Points:**

* The paper has formal depth and looks well thought through
* The language is good
* The approach is useful and novel

**Subreviewer:**

I submitted this review.

**Weak Points:**

* The contributions of the paper are unclear. SMOL (syntax+semantics), the ontology for describing executions, the formal mapping, the querying? What of those is the main topic of the paper? I would recommend the authors invest some time in better structuring the paper.
* The authors use terms as if they were established technical terms ("semantic debugging", "semantic programming"), but I would not know their meaning, Google didn't help
* I would expect a dedicated related work section with a thorough discussion of more than one paper, not a subsection to the conclusion and some pointers in the introduction
* As the authors have a running implementation, I would be interested if the authors have some motivation from application
* With ESWC being a semantic web venue, I could imagine that some brief introduction into the relevant terms and challenges from computer programming would be useful

Minor:
* "a simple while language for statements" (p.4) sounds awkward

---

> ### Author Rebuttal · Authors · 2021-01-29
>
> # Answer to AnonReviewer 3:
>
> We thank the reviewer for the concrete pointers to additional related papers. We agree with these remarks and will add a separate section on related work in the final version of the paper
>
> ## On the contribution:
> The contribution of this work is three-fold (1) the concept of interpreting a program state as a knowledge graph to enable semantic state access (2) the application enabled by this concepts and (3) the SMOL language that implements the concept and exemplifies the applications. We will clarify this better in the intorduction.
>
> We will make our position w.r.t. to [1-3] more explicit:
> Contrary to [1] and [2] we are not investigating programming languages that operate directly on DL interpretations or knowledge graphs, but
> investigate how programming languages can be enhanced with semantic technologies. Contrary to [3] we do so not on external knowledge graphs, but on an internal one.
>
>
>
> ## On motivation and targeted applications
> We are motivated by our experiences with formalizing evolving systems and developing simulators (such as Railway systems [6]) where
> the resulting applications duplicate domain knowledge in both data structures and behavior.
> For such applications, which are very near to their domain, it is difficult to (a) spot semantic bugs, which are only recognized as bugs by domain experts
> (b) reduce redundancy w.r.t. established ontologies, as the program must not only mirror the application domain, but also follow established practices to be well-simulatable.
> Semantic programming and debugging allow us to solve these problems.
>
> As mentioned in the article, we have re-implemented one such system for geological processes [11] in SMOL. We will add a more detailed discussion on such systems to the introduction. We hesitate to fully include this case study, as introducing the geological domain seems beyond the scope for this paper and would obscure the conceptual contribution.
>
> ## Specific points:
>  - We are aware that the interpretation of program states as FO model is not novel, it is this very interpretation that is behind the given proposition.
>  - Semantic debugging and semantic programming are terms introduced in this paper.
>  - We thank the reviewer for the suggestion for a subsection that introduces the relevant PL terms to the SW audience. As this introduction and the additional discussion of related work may take up some place, we plan to publish a technical report that contains the full version of 3.1 and remove the transition system from the conference publication, if necessary.

---

### Official Review · AnonReviewer5 · 2021-01-17
**This paper proposes the notion of semantic programming, combining domain knowledge and rules to create domain-independent programs that can be validated depending on different domains.**

**Rating:** 1
**Confidence:** 3
**Impact:** 2
**Design And Technical Quality:** 3

**Review:**

This paper proposes the notion of semantic programming, combining domain knowledge and rules to create domain-independent programs that can be validated depending on different domains.

The paper is well written, but I found it very hard to follow due to the amount of acronyms used in the publication and the examples chosen to illustrate the approach. I think there are nice ideas behind this work and the approach is quite novel, but the status of the work may be too preliminar to be presented at ESWC. I provide more details for my rationale below.

- The motivation and examples chosen are not convincing: Adding a layer of semantics to programming may be an excellent idea to validate programs prior to their execution, automatically generate code aligned with knowledge graphs, generate self describing traces, etc. At the same time, it also introduces complexity, taking into account that most developers are foreign to standards such as OWL and SPARQl. For this reason, stating that a car could be running with a wheel makes me ask: Why couldn't we make sure that in the Car class we require all these parameters to be present? Isn't an additional validation in the code simpler? Similarly, with the tree example, why can't we have different classes to represent the types of nodes?

- Part of my confusion comes from some of the examples themselves, I believe they should be improved to emphasize the benefits of the approach. For example, in Figure 5 there are some arrows that I don't understand: What is the relationship from p1 and X? Is it o? is it runsMethod? In the arrow from X to 2, what is f? Why is it in the middle of the arrow as well, does it have a meaning, or is it just the way the figure is arranged?

- In Fig 8, the authors emphasize that the function being returned is not an arithmetic function. At the same time, the final value returned by the method is "return s*this.w*this.h", which is indeed a product. Even if the object sent to the the Rectangle class is "scene", why not get the scale associated with the scene directly in the method?

- Just from the graph generated in Figure 5, I wonder about the plausibility of the approach. Sure, the authors show that hundreds of elements may be added to the graph and it still resolves in a reasonable time, but having to wait several seconds when debugging doesn't sound appealing for programmers. How much are the authors expecting to record when doing these kind of validations and inferences?

- The evaluation only shows the feasibility of the approach, it doesn't really answer:
  1) Would developers be comfortable with the proposed system? Would they be able to use it, or is a knowledge engineer needed in this process?
  2) Does the system scale beyond the sample use cases selected?
  3) What is the level of effort needed to implement the approach versus adapting code? (as it needs rules/shapes and domain knowledge explicitly represented)

**Anonymity:**

No, I would like my review to be deanonymized.

**Reuse And Availability:**

3: Medium

**Strong Points:**

- Novel approach that mixes traditional programming with semantic web technologies.
- The approach has potential benefits for validation and embedding semantics in code
- An implementation is provided

**Subreviewer:**

I submitted this review.

**Weak Points:**

- The paper is hard to follow.
- The examples chosen do not showcase the main benefits of the approach convincingly
- The evaluation is preliminary
- Although an implementation is provided, it doesn't seem documented enough for usage. For instance, the examples are barely described, the readme just mentions how to run them and their contents.

---

> ### Author Rebuttal · Authors · 2021-01-29
>
> # Answer to AnonReviewer 5
> We thank the reviewer for the very constructive and thorough comments.
>
> ## On the motivating example:
> Concerning alternative modeling approaches: The nodes of a 2-3 tree change their "class" during their lifetime from leaf to 2-node to 3-node.
> It would be highly inefficient to create a new object every time this happens, as the addition of one value to the tree may cause several such changes.
> This exemplifies our above point where the application "domain" (here: 2-3 trees) and the needs of the implementing language (here: efficiency) collide.
> We are going to clarify this point at the end of Ex. 1.
>
>
> ## On Figure 5:
> All fields are subclasses and subproperties of Field. Analogously for variables.
> By being properties, we can express that to use them to connect an object O1 with the object O2 stored in its field F using the triple F(O1, O2).
> By being objects, we can express that they belong to a class. E.g., if O1 is of class C using the triple hasField(C, F).
> This is also the reason the f occurs both over an arrow (from X to 2) and at the end of one in the figure (hasField).
> p1 is connected to X using the variable o.
>
> We will mark the nodes in Fig. 5 according to their layer (syntactical/object/process) and rearrange the labels.
>
>
> ## On Figure 8:
> When we say that the method is "not merely an arithmetic expression", we mean that the whole method contains additional, non-arithmetic expressions (namely, a method call). Only the final computation is arithmetic. We will clarify this in the text.
>
>
> ## On overhead for the developer:
> The mapping and the ontology has to be performed only *once* per language, as part of the vocabulary (such as class names, methods, etc.) are extracted from a program and requires no additional input from the user. Semantic debugging requires, thus, little effort once the initial setup is done.
>
>
> ## On scalability:
> The current state of the implementation is explicitly not optimized for speed.  We expect the approach to scale both to bigger states (as discussed, we are investigating virtual graphs for this purpose) and larger program languages. Concerning larger programming languages we expect that applying SSA on a more complex language will provide even more benefit, as their states are more complex and harder to investigate in detail. This is in particular the case for languages such as Java, where the JVM implementation of the language drastically differs from the surface syntax exposed to the programmer. An ontology for a mainstream language, however, is beyond the scope for this paper and would obscure the conceptual contribution.

---

> > ### Comment · AnonReviewer5 · 2021-02-02
> > **Thanks for your answers**
> >
> > I would like to thank the authors for clarifying my concerns. I am a little confused by one aspect though. An element usually cannot be both a class and a property, as it would mess with the RDF metamodel, and would be extremely confusing. For example, that would mean that a property could have as domain another property and be subclass of another class. Given that this is a Semantic Web conference, I find that this is an important issue.
> >
> > On another note, I think the example is still not selling the idea quite convincingly, as a class changing its type looks like an outlier rather than something developers would encounter frequently. Hopefully the clarifications from the authors will help.
> >
> > I have decided to raise my final score, but I am still divided on whether the paper should be accepted at ESWC.

---

### Decision · Program_Chairs · 2021-02-23

**Decision:**

Accept with shepherding

**Comment:**

The key positive aspect of the paper is in addressing a novel problem (at least for the application of semantics) and proposing an interesting/original approach to it. This was an aspect highly appreciated by all reviewers.  However, this positive aspect is overshadowed by a number of  concerns: (1) the novelty is not clear, partially because related work is weakly covered; (2) the evaluation is preliminary in does not allow assessing whether the approach is scalable/generalizable/useful in practice; (3) the paper is hard to follow, also possibly because the terminology was not sufficiently well explained for the SW-community.

The authors submitted an extensive rebuttal, which was appreciated by the reviewers and addressed some concerns. Post-rebuttal, the paper has been discussed intensively in the PC, since, despite it's limitations,  the papers idea was perceived as very novel. Considering that the KG track explicitly called from experiences from other communities than the Semantic Web core community with Knowledge Graphs, it was decided to conditionally accept this paper through shepherding.

In particular, authors are asked to address concerns (1) and (3) in the newly submitted version, as well as all other comments of the reviewers that can be feasibly addressed within the next weeks. A final decision will then be reached based on this new version of the paper.